# Exploring thematic structure and predicted functionality of 16S rRNA amplicon data

**Stephen Woloszynek**[ID][1]*, **Joshua Chang Mell**[2], **Zhengqiao Zhao**[1], **Gideon Simpson**[3], **Michael P. O'Connor**[4], **Gail L. Rosen**[1]*

**1** Department of Electrical and Computer Engineering, Drexel University, Philadelphia, Pennsylvania, United States of America, **2** Department of Microbiology and Immunology, Drexel University College of Medicine, Philadelphia, Pennsylvania, United States of America, **3** Department of Mathematics, Drexel University, Philadelphia, Pennsylvania, United States of America, **4** Department of Biodiversity, Earth, and Environmental Science, Drexel University, Philadelphia, Pennsylvania, United States of America

* sw424@drexel.edu (SW); glr26@drexel.edu (GLR)

**Data Availability Statement:** All code for analyses is available on github at https://github.com/EESI/exploring_thematic_structure. The package is available on CRAN at

## Abstract

Analysis of microbiome data involves identifying co-occurring groups of taxa associated with sample features of interest (e.g., disease state). Elucidating such relations is often difficult as microbiome data are compositional, sparse, and have high dimensionality. Also, the configuration of co-occurring taxa may represent overlapping subcommunities that contribute to sample characteristics such as host status. Preserving the configuration of co-occurring microbes rather than detecting specific indicator species is more likely to facilitate biologically meaningful interpretations. Additionally, analyses that use taxonomic relative abundances to predict the abundances of different gene functions aggregate predicted functional profiles across taxa. This precludes straightforward identification of predicted functional components associated with subsets of co-occurring taxa. We provide an approach to explore co-occurring taxa using "topics" generated via a topic model and link these topics to specific sample features (e.g., disease state). Rather than inferring predicted functional content based on overall taxonomic relative abundances, we instead focus on inference of functional content within topics, which we parse by estimating interactions between topics and pathways through a multilevel, fully Bayesian regression model. We apply our methods to three publicly available 16S amplicon sequencing datasets: an inflammatory bowel disease dataset, an oral cancer dataset, and a time-series dataset. Using our topic model approach to uncover latent structure in 16S rRNA amplicon surveys, investigators can (1) capture groups of co-occurring taxa termed topics; (2) uncover within-topic functional potential; (3) link taxa co-occurrence, gene function, and environmental/host features; and (4) explore the way in which sets of co-occurring taxa behave and evolve over time. These methods have been implemented in a freely available R package: https://cran.r-project.org/package=themetagenomics, https://github.com/EESI/themetagenomics.

https://CRAN.R-project.org/package=themetagenomics. The developmental version of the package and code is located at https://github.com/EESI/themetagenomics. Data are already publicly available; accession numbers are provided in manuscript.

**Funding:** This work was supported by the National Science Foundation under grant #1120622 to GLR. The funder had no role in study design, data collection and analysis, decision to publish, or preparation of the manuscript.

**Competing interests:** The authors have declared that no competing interests exist.

## Introduction

High-throughput sequencing now permits for the analysis of multiple large datasets on the microbiome and diseases of interest. Historically, researchers have sought to reduce the dimensionality of the data and/or perform feature selection to identify species (or other taxa) of interest that are correlated with sample/community-level attributes (which we will refer to as "phenotypic" attributes or "phenotypes") like host health status. Unfortunately, these phenotype-associated species may co-occur with the same or different proportions across samples within the same phenotype. Capturing these configurations is of interest to us, as we contend it is more informative than merely finding specific taxa [1,2].

Nevertheless, obtaining meaningful configurations or subsets of taxa is often a daunting task. These high-dimensional microbiome datasets include categorical and numeric features associated with each sample. These, in turn, may be linked to a set of taxonomic abundances that are derived from clustering similar sequencing reads. Typically, taxonomic markers, such as variable regions of the 16S rRNA gene common to all prokaryotes, are used to perform the clustering based on a fixed degree of sequence similarity among reads. Such clusters are termed Operational Taxonomic Units (OTUs), and each OTU is usually assigned to some level of taxonomy, such as a genus. Identifying OTUs correlating with specific sample features (e.g., body site, disease presence, diet, age) can be done via unsupervised exploratory methods [3]. Unfortunately, complexities inherent to taxonomic abundance data hinders many of these methods. These complexities include vastly more OTUs relative to the number of available samples [4], substantial sparsity in the OTU counts (absence of organisms in most samples), and differences in sampling depth among samples. The sampling depth issue then requires normalization, introducing additional challenges. In particular, the normalization transforms the abundances into relative abundances within each sample (compositional data) [5,6]. Common approaches (e.g., differential abundance analysis [3,7,8] and regularized regression [9,10]) associate indicator taxa with sample information, leading to overly simplified biological interpretations.

From an ecological perspective, co-occurring OTUs may represent related subcommunities of taxa, which consist of OTUs that are common to (or overlap with) each sample. This overlap is due to taxa that covary with host or environmental factors; thus, identifying important subcommunities (groups of taxa) and configurations of taxa (the grouping and ratios/relative abundances of co-occurring taxa) may allow for a more biologically meaningful interpretation than identifying indicator OTUs, because identifying subcommunities preserves the groupings and abundances of taxa [2,11–13]. Developing techniques for identifying subcommunities is a fundamental goal of this work.

Methods that predict functional profiles from 16S rRNA survey data usually report the overall function of a sample and do not provide granularity on how each subcommunity provides specific functions (Fig 1). Standard methods that predict function from 16S rRNA survey data include PICRUSt, Tax4fun, Piphillin, and SINAPS [14–17]. These simulate gene abundances from the OTU relative abundance profile by assigning pre-existing gene ontologies, based on whole genome sequences, to the OTUs. The simulation is trivial for known microbes, but for novel OTUs, gene content is interpolated through its neighbors' genes. These are determined via an unsupervised phylogenetic tree reconstruction. However, after the gene abundance profiles are simulated for an entire sample, a user cannot view which functional content associates with which taxa, nor how subcommunities contribute to function.

We consequently have developed themetagenomics, a novel pipeline for analyzing 16S rRNA amplicon surveys that (1) identifies subcommunities associated with specific sample features and (2) uncovers functional profiles that further characterize these subcommunities. We

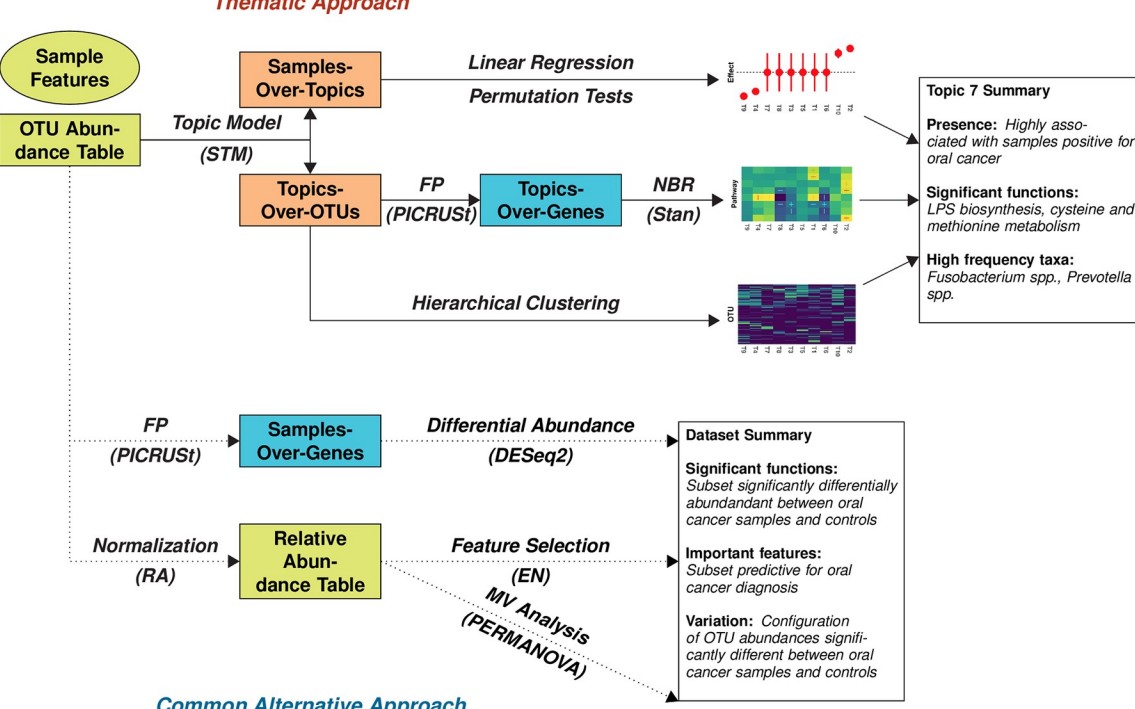

**Fig 1.** (Thematic Approach) Given a 16S rRNA gene abundance table, a topic model is used to uncover the thematic structure of the data in the form of two latent distributions: The samples-over-topics frequencies and the topics-over-OTUs frequencies. The samples-over-topics frequencies are regressed against sample features of interest to identify the strength of a topic-covariate relationship to rank topics (top). The topics-over-OTUs frequencies are used in a gene function prediction (FP) algorithm to predict gene content. Important functional categories are identified via a fully Bayesian multilevel negative binomial (NBR) regression model (middle). The topics-over-OTUs distribution is hierarchically clustered to infer relationships between clusters of co-occurring OTUs and topics (bottom). The result is the ability to identify key topics that associate clusters of bacteria and their associated functional content to sample information of interest. (Alternative Approach). A common alternative approach currently used in the literature involves independently (1) characterizing the taxonomic configuration and (2) predicting the functional configuration of the OTU abundance table. Gene function prediction is performed on the full OTU abundance table, followed by a differential abundance analysis to infer differences in specific genes between sample features of interest (top). The OTU table is normalized to overcome library size inconsistencies and then analyzed via two methods: (1) an elastic net (EN) to find sparse sets of OTUs that are predictive for the sample feature of interest (middle) and (2) a multivariate (MV) analysis to identify relationships between beta diversity and the sample feature of interest (bottom). The result are three analyses that summarize the entire OTU relative abundance table, unlike the thematic approach, which characterizes co-occurring sets of OTUs (configurations) in three ways.

use a topic model approach to uncover subcommunity structure by estimating taxonomic co-occurrence. Topic models are dimensionality reduction techniques that have had considerable use in natural language processing to represent, as topics, co-occurrence relationships between words from a corpus of documents. They have more recently shown promise as a method for exploring taxonomic abundance data [2,18], where topics act as low-dimensional representations of co-occurring sets of taxa given a set of samples, i.e., far fewer topics than OTUs (Table 1). Unlike other dimensional reduction techniques common to microbiome data analysis (e.g., principal coordinate analysis), topic models provide a new set of features (topics) that should be familiar to microbiome researchers in that they have a form similar to relative abundances: each sample is represented as a vector of frequencies across topics and each topic is represented as a vector of frequencies across taxa. Lower dimensional features that are also familiar may ease their interpretation.

Our pipeline aims to concisely summarize high-dimensional data in the form of OTU abundances as low-dimensional sets of co-occurring taxa (topics) with their corresponding

**Table 1. Relationship of terms.**

| Topic Model | Pipeline | Description |
|---|---|---|
| Document | Sample | Collection of reads from subject $m$ at time $t$ |
| Topic | Topic | Collection of co-occurring taxa, subcommunity |
| Word | OTU, Gene, Taxa | Features from taxonomic abundance table or predicted functional content |
| Document-Level Covariate | Sample information, Sample class | Sample-level variable of interest–e.g., disease presence, diet, rainfall, time |
| θ | Samples-Over-Topics Distribution | Vector of topic frequencies in a given sample; probability of a topic occurring in a given sample |
| β | Topics-Over-OTUs Distribution | Vector of OTU frequencies in a given topic; probability of an OTU occurring in a given topic |

predicted functional potential. When additional high-dimensional data is available (e.g., predicted gene function abundances), interpretability becomes increasingly difficult. Although topic models have been applied to microbiome data because of their interpretable features, no work has been done to leverage their interpretability to link low-dimensional representations of OTU and predicted gene function abundances. In addition, little research addresses ways to fully leverage the latent features topic models extract from microbiome data. For example, correlated topic models [19] not only capture taxonomic co-occurrence but also topic co-occurrence, such that the frequency of two topics, with different sets of co-occurring taxa, occurring in any given sample, may be positively correlated. This is the basis of our novel approach to exploit the correlation structure of topics across samples to resolve long-term temporal behavior of subcommunities (represented as topics) in microbiome time-series datasets.

Our approach at linking taxonomic composition to predicted functional content (obtained via methods that leverage preexisting gene ontologies) within topics is unique. We apply a recently developed structural topic model (STM) [20] to a novel domain (16S rRNA amplicon surveys), where each topic represents a cluster of co-occurring OTUs and each OTU can occur in multiple topics with varying frequency. Functional content is then predicted within-topic, allowing the topics to act as low-dimensional taxonomic and functional summaries of the input data. The topics are then linked to sample-information that reflects host or environment status. Topics-of-interest (e.g., those that contain differentially-enriched functional profiles) can easily be identified in our pipeline via a fully Bayesian multilevel regression model. We also apply our approach to empirical time-series data where we characterized events in terms of sets of correlated topics to explore how the taxonomic configurations evolved over time.

Our pipeline has been implemented in the R package themetagenomics: https://cran.r-project.org/package=themetagenomics, https://github.com/EESI/themetagenomics.

## Results and discussion

Here we explore the use of themetagenomics on publicly available datasets studying Crohn's disease microbiota (Gevers et al. [21]), oral cancer microbiota (Schmidt et al. [22]), and the variation of microbiota as a function of time (David et al. [23]). With the larger Gevers et al. Crohn's dataset, we validate the ability of themetagenomics to capture microbial profile "signatures" (configurations of taxa which are groups with specific ratios/relative abundances of co-occurring taxa). We show that (1) topics generalize well to test data not initially seen by the model (generalizable topics are topics robust to overfitting, such that they avoid fitting noise and thus can capture important signals representative of true taxonomic co-occurrence

profiles), and (2) topics capture distinct microbial signatures found in the original OTU relative abundance data.

After validating the configuration of taxa within-topic (by assessing classification performance to evaluate topic generalizability and OTU co-occurrence to evaluate topic quality) and the configuration of predicted gene functions within-topic (via a permutation test using metagenomic data), we assess the biological relevance of our low-dimensional summaries (topics). We then apply our complete pipeline to Gevers et al. [21] to link a topic's functional content, taxonomic co-occurrence, and sample information (clinical diagnosis of Crohn's disease (CD)), and we compare these results to those obtained by the original authors. We compare our results to those obtained by DESeq2 and an alternative topic-model based microbiome analysis tool, BioMiCo [2]. We validate the functional prediction of our pipeline with the oral cancer Schmidt et al. [22] dataset by showing the low-dimensional topic profiles identified by themetagenomics are also present in complementary metagenomic shotgun (MGS) sequence data. We lastly implement our approach on time-series gut microbiome data from David et al. [23]. We interpret the results in terms of topics and posterior uncertainty and compare our findings to those obtained by a HC approach, as well as the results reported by David et al. [23].

## Topic modeling Feasibility and generalizability

We assess (1) if topics correlate to sample phenotypes (e.g., disease state) and (2) whether those topics generalize well–that is, can the learned topics predict phenotypes from new data. Using a random forest classifier, we compared the classification performance between two different sets of predictors: (1) frequencies of topics-across-samples, θ, from the STM, and (2) OTU relative abundances across samples generated from QIIME [24]. For this analysis, we focused on the Crohn's disease study from Gevers et al. [21] given its large sample size (555 terminal ileum samples).

To assess generalizability, we used a training/testing approach. We randomly selected 80% of samples as our training set; the remaining 20% were set aside for testing (Table A in S1 File). Class labels were binary, with positive (CD+) and negative (CD-) clinical diagnoses acting as the positive and negative classes, respectively. For classifying CD diagnosis, we hypothesized that using topics as predictors would outperform using relative abundances of OTUs, since the relative abundance-based predictors are sparser, whereas topic modeling performs dimensionality reduction, resulting in a relatively smaller set of topics that are less sparse relative to OTUs. There was little difference between the topic model with at least 25 topics and the OTU table to train the classifier (Fig A and Table B in S1 File). During testing, however, using topics as features outperformed relative abundances, particularly in the F1 score, with relative abundances achieving 80.8% and at least 25 topics achieving greater than 82.1% (Table C in S1 File). Using OTU relative abundances as predictive features resulted in a larger proportion of false negatives, which was likely due to its reliance on few, relatively rare taxa. Topics, on the other hand, are less reliant on rare taxa because dimensionality reduction generates less sparse features (S1 Appendix).

## Correlation between topics and phenotype

To identify topics of interest that were strongly associated with phenotype, we again implemented themetagenomics on the Crohn's disease dataset, using the same binary indicator for CD diagnosis as above. We then performed posterior inference. The primary output of the topic model, as with any Bayesian analysis, is a posterior distribution of quantities that estimate latent variables-of-interest (e.g., the frequencies of topics, θ, in a particular sample) given the

observed data (e.g., OTU abundances). Posterior inference involves sampling these latent variables-of-interest from the posterior distribution of the fitted topic model to calculate expected means and assess uncertainty in those expectations.

With the posterior distribution, we identified topics-of-interest based on their "topic-sample-effects"–the regression coefficients that represent differences in topic frequencies between CD+ and CD- samples. We performed permutation tests to ensure that detected topic-sample-effects were not spurious (S1 Appendix). For the model with 25 topics (K25), we performed 25 permutations, where we randomly permuted class label assignments (CD+, CD-), refit the topic model, and calculated the mean regression coefficient for each topic. Of the 25 topics, 8 topics had 95% uncertainty intervals for the effect size (differences between CD+ and CD-) that did not span 0 (Figure B in S1 File). We consider these "high-ranking-topics." Topics T15, T12, T2, and T14 had estimates greater than 0 (implying robust associations with CD+), whereas topics T11, T25, T13, and T19 had estimates less than 0 (implying robust associations with CD-). Increasing the number of fitted topics gave similar results; for K75, 14 topics did not span 0 (Figure C in S1 File).

We next tested how well a topic model (fit with the binary CD encoding) could capture the severity of disease using the Pediatric Crohn's Disease Activity Index (PCDAI) associated with CD+ that increases as CD severity increases (CD- samples were set to PCDAI = 0). The frequency of a sample containing a particular topic given its PCDAI is shown in Fig 2A for models K25 and K75. Topics are color-coded based on their association with CD, which is estimated using their topic-sample-effects (yellow and violet represent topics most and least associated with CD, respectively). Each overlapping line represents one of 25 replicate simulations.

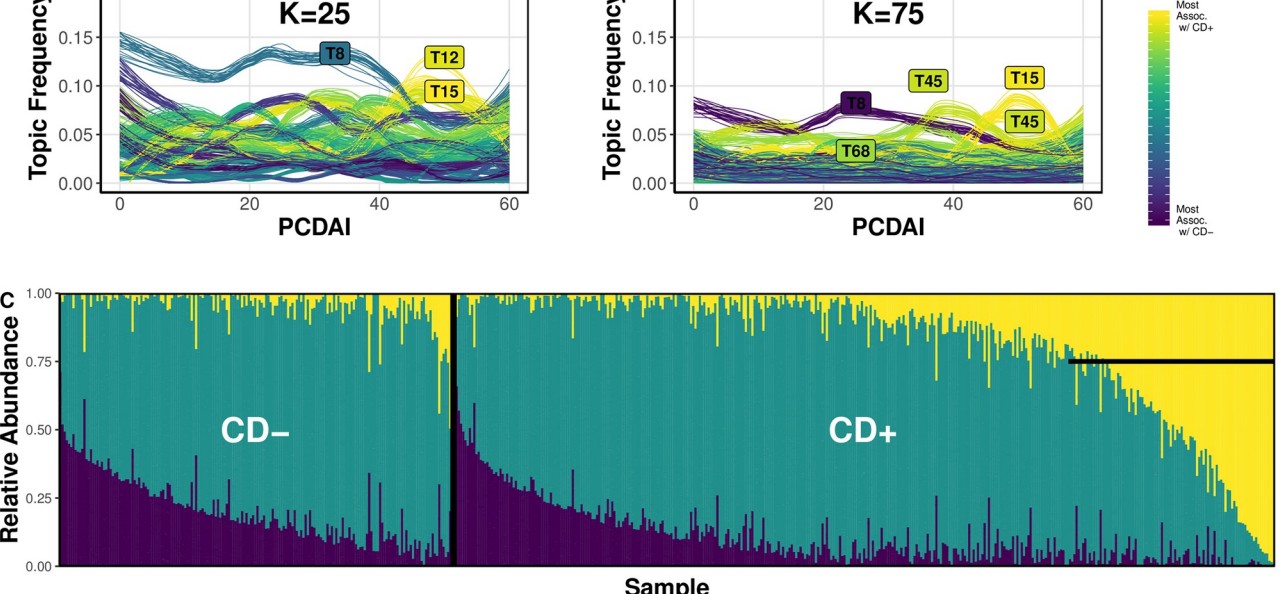

**Fig 2.** (A). The relationship between topic frequency within a sample and that sample's Crohn's Disease (CD) severity (PCDAI score) for the 25-topic STM. Each line represents the frequency of a topic as a function of sample PCDAI score. High frequency topics are labeled. Violet and yellow color-coded trajectories designate CD- and CD+ associated topics, respectively. Posterior sampling was performed across 25 replicates, with each line plotted to represent the distribution of the topic frequency trajectories. (B). Trajectories for the 75-topic model. (C). The relative abundance of OTUs in the (input) OTU relative abundance table for "noteworthy" OTUs from high-ranking-topics. The left and right panels show the relative abundance of these OTUs in each CD- and CD+ sample, respectively. Noteworthy OTUs are defined as high-frequency OTUs, sampled from the posterior distribution, that concentrate into high-ranking-topics (yellow = CD+ topic group, violet = CD- topic group, green = unassociated topic group). The horizontal line marks a subset of samples that contain a large proportion of the OTU profile associated with CD+ high-ranking-topics.

Both panels demonstrate that as PCDAI increases, the thematic profile shifts from one dominated by a single CD- associated topic (T8) to a set of CD+ topics (T12, T15, T45). The transition occurs at approximately PCDAI = 35. Because the K25 model had greater separation of high probability topics, it will be the focus for the remainder of analyses involving Gevers et al. [21] data.

From the posterior topics-over-OTUs distribution (β) for the K25 model, we identified OTUs highly associated with CD, that is, OTUs with high frequency in high-ranking-topics (CD+ associated topics T19, T13, T25, T11; CD- associated topics T14, T2, T12, T15) in more than 99% of posterior samples (arbitrary threshold). We categorized these OTUs as CD+ associated OTUs, CD- associated OTUs, and unassociated OTUs. Fig 2C shows the relative abundances of the 3 groups for each sample in the QIIME-generated OTU abundance table. Of CD+ samples (right of vertical black bar), approximately 25% were characterized by a greater proportion of CD+ associated OTUs relative to CD- (marked by the horizontal black bar). The ratio of CD- associated OTUs to unassociated OTUs had a similar distribution among CD+ and CD- samples, suggesting that the OTU profile from CD+ high-ranking-topics is specific for the CD+ disease status. Lastly, when we regressed PCDAI against the relative abundances of the CD+ associated OTU profile, we found a significant positive relationship (β = 0.057, p = 0.01, 100 permutations), albeit explanatory for only a small portion of the variation ($R^2$ = 8.64%), suggesting that presence of this OTU profile may be weakly indicative of severe cases of CD (S1 Appendix).

**Comparison to BioMiCo.**   We compared our approach's performance to BioMiCo, a topic model that identifies meaningful sets of "assemblages" (analogous to topics–i.e., sets of cooccurring taxa) by directly incorporating sample- or environmental level features (labels) during the training procedure. It is fully supervised and assumes that a sample is comprised of a mixture of communities that share sample- or environmental level features. These communities are described by a set of high probability assemblages which are in turn described by a set of high probability taxa.

We fit BioMiCo using 25 and 50 assemblages and compared its ability to distinguish CD from control using held-out testing data (same train/test splits as described previously) and then compared these results to the prediction performance of the STM. Testing performance was similar between the two approaches (Table C and Table F in S1 File). The balanced accuracy was highest for the 25-topic STM model, but the STM's performance varied as a function of topic number. F1 score, however, was much worse for BioMiCo due to its low precision.

For the 25-assemblage model, there were roughly four assemblages with high posterior probability for CD samples and low posterior probability for controls. If we focused on the taxa with the top-10 highest posterior probability of belonging to these assemblages, no more than 2 taxa were present in the top-10 highest probability taxa in the STM's CD-topics that were most associated with CD, suggesting little correspondence between the composition of assemblages and topics. Alternatively, when focusing on assemblages with high posterior probability for control but not CD, one assemblage had 4 genera in common with the STM's topic 13: *Parabacteroides*, *Bacteroides*, *Ruminoccous*, and *Roseburia*.

It is worth noting, however, that the STM and BioMiCo aim to characterize data differently and hence the distribution of taxa within a given topic are expected to be different. Still, both approaches show they similarly generalize to new data. An advantage of themetagenomics is that it leverages output inherent to the design of the STM that is not available via BioMiCo, notably topic-topic correlation. Also, the STM is appreciably faster, taking minutes to run on the Gevers data whereas BioMiCo took days. Unlike BioMiCo–as well as the STM which is aimed for more general use–themetagenomics delivers a framework that facilitates ease-of-use microbiome analysis using a topic model via an R package with a variety of intuitive functions

for preprocessing, analyses, and visualizations. It also provides novel downstream approaches such as time series analysis which leverages the STM's estimation of topic-topic correlation, as well as methods to associate a topic's taxonomic composition to its predicted gene functions.

## Linking function to taxonomy with topics

We wanted to discern whether the topics would continue to identify meaningful relationships upon introducing another layer of information: predicted function (via abundances of metabolic pathways). Consequently, we applied our full themetagenomics pipeline to the Crohn's disease dataset and compared our findings to those of the original authors. To further characterize topics, we applied PICRUSt to the topics-over-OTUs distribution, β, to predict the functional gene content within topics. The genes were then annotated in terms of their KEGG functional hierarchy designation [25], thereby providing each gene with a metabolic pathway label. We then performed a fully Bayesian multilevel regression analysis on the predicted abundances of each gene to identify strong topic-pathway interactions.

Like Gevers et al. [21], we identified an increase in membrane transport associated with CD+ subjects' gut microbiome; however, using themetagenomics, we were able to pinpoint the specific topics associated with the enrichment of these functional categories, T2 and T12 (Fig 3A). We then could link enrichment of membrane transport genes to the taxa that were also enriched in this topic. For example, topics T2 and T12 were dominated by Enterobacteriaceae. These Enterobacteriaceae-enriched topics were also enriched for siderophore and secretion system related genes. Like T2 and T12, T15 was highly associated with CD+; however, it was less enriched for membrane transport genes. This suggests that the cluster of bacteria found in T15 (*Haemophilus* spp., *Neisseria*, and *Fusobacteria*) may have contributed less to the shift of transport genes reported by Gevers et al. [21] and instead have distinct pathway associations with CD.

The strongest topic-pathway interaction was found in T19 for genes encoding bacterial motility proteins. For T19, three motility-related pathways (bacterial motility proteins, bacterial chemotaxis, flagellar assembly) had topic-pathway interactions that did not span 0 at 80% uncertainty, suggesting that T19 was more enriched in cell motility genes relative to all other topics. The pathways inferred from T19 are consistent with this taxonomic profile, which consisted of motile bacteria belonging to Lachnospiraceae, Roseburia, and Clostridiales. Enrichment of two lipopolysaccharide (LPS) synthesis categories were associated with CD+ topics; however, one of these categories was specific for only T15 (Table D in S1 File).

**Comparison to DeSeq2.** We compared the topics' functional profiles to the results obtained by performing a DESeq2 differential abundance analysis on functional predictions obtained by applying PICRUSt to the QIIME-generated OTU abundance table. Of the 160 (level-3) KEGG pathway categories, more than half (87) were found significant ($\alpha < 0.1$) in the DESeq2 approach, despite using Bonferoni correction (a conservative approach to correct for multiple comparisons), complicating interpretation (Fig 3B). Despite minor differences in specific pathway enrichment between themetagenomics and the DESeq2 approach (S1 Appendix) the major difference was the greater number of low-uncertainty/significant pathway categories found by DESeq2. While one could reduce the significance level when applying DESeq2 to achieve a smaller subset of significant pathway categories, the choice is arbitrary. Moreover, the predicted functional abundances (via PICRUSt, Tax4fun, etc.) are scaled based on the abundance of taxa from which they were derived. Thus, high taxonomic abundances will often yield high functional abundances. Many of the significant pathway categories identified by DESeq2 may be driven by a small subset of highly abundant taxa. Themetagenomics, on the other hand, first groups co-occurring taxa into topics. Because functional prediction is performed within a

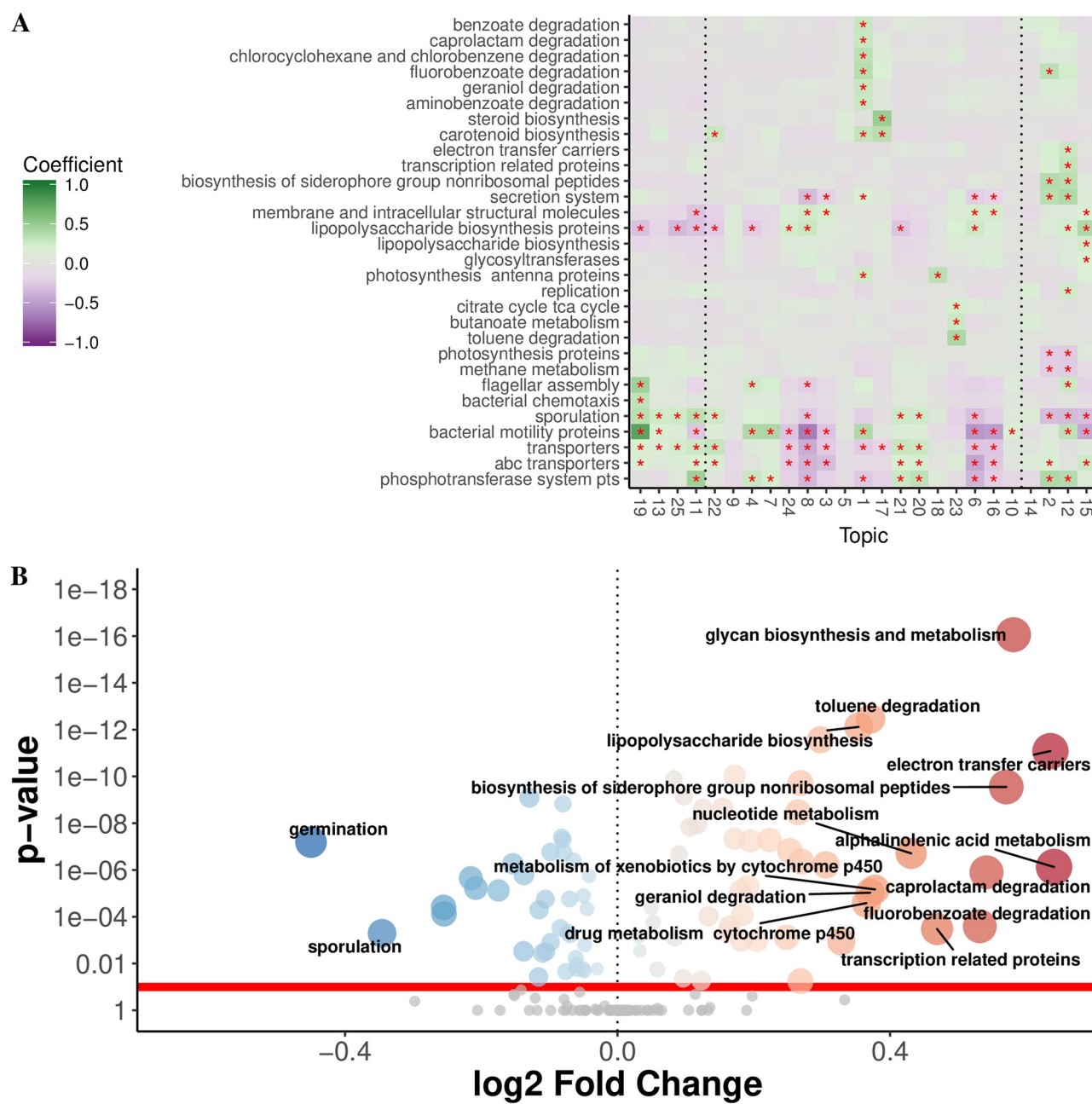

**Fig 3.** (A). Level-3 pathway category-topic interaction regression coefficients from the multiple level negative binomial model. Red asterisks indicate estimated pathway-topic interaction weights that do not span 0 at 80% uncertainty (pathways lacking robust interactions are omitted). Green = large positive coefficients thus enrichment for that pathway in that topic, Violet = large negative coefficients thus depletion for that pathway in that topic. Topics are ordered from CD- associated (left, T19) to CD+ associated (right, T15). High-ranking-topics are delineated by the vertical dotted lines (CD-: T19-T11; CD+: T14-T15). (B). Volcano plot showing DESeq2 results for differentially abundant predicted level-3 KEGG categories. Functions were predicted using PICRUSt on the copy number normalized OTU abundance table. Blue points represent categories significantly enriched for CD- and red points are categories enriched for CD+, respectively. Gray points are categories with p-values greater than 0.1 after Bonferroni correction.

topic, taxa that are highly abundant in the input OTU abundance table can only affect the topics in which they are present at high frequency. Thus, this prevents high abundance taxa associated with a subset of samples (e.g., CD+), and their corresponding predicted pathway abundances, from disproportionately influencing the statistical significance of these pathways.

## Validating the functional predictions of themetagenomics via Paired MGS samples

Using sample-matched (N = 12) oral cancer microbiome samples from Schmidt et al. [22] that underwent both 16S rRNA amplicon sequencing and metagenomic shotgun sequencing, we verified enrichment or depletion of predicted functional content (collapsed into metabolic pathway categories) of the themetagenomics pipeline. The pipeline processed the 16S rRNA samples and compared the results to metagenome-based gene functional abundance data. Fig 4A shows the relative enrichment/depletion of various topic-pathway combinations identified by themetagenomics. For example, bacterial motility genes were enriched in topic 25 (positive coefficient, shaded green), whereas bacterial motility genes were depleted in topics 3 and 9 (negative coefficients, shaded violet).

To compare the results from themetagenomics to gene function abundances inferred from metagenomic shotgun sequencing for each topic, we first identified high frequency taxa (those with frequencies greater than 1% in that topic) then identified all reads belonging to these taxa in the metagenomic shotgun data. To identify pathway-topic enrichment/depletion, we then applied a multilevel regression model. The results indicate that the taxa belonging to a topic are associated with an enrichment/depletion of genes present in the shotgun data (Fig 4B). Notably, LPS biosynthesis proteins and porphyrin metabolism pathways were depleted in multiple topics in both sets of results. The relative enrichment/depletion of phosphotransferase system genes was also similar.

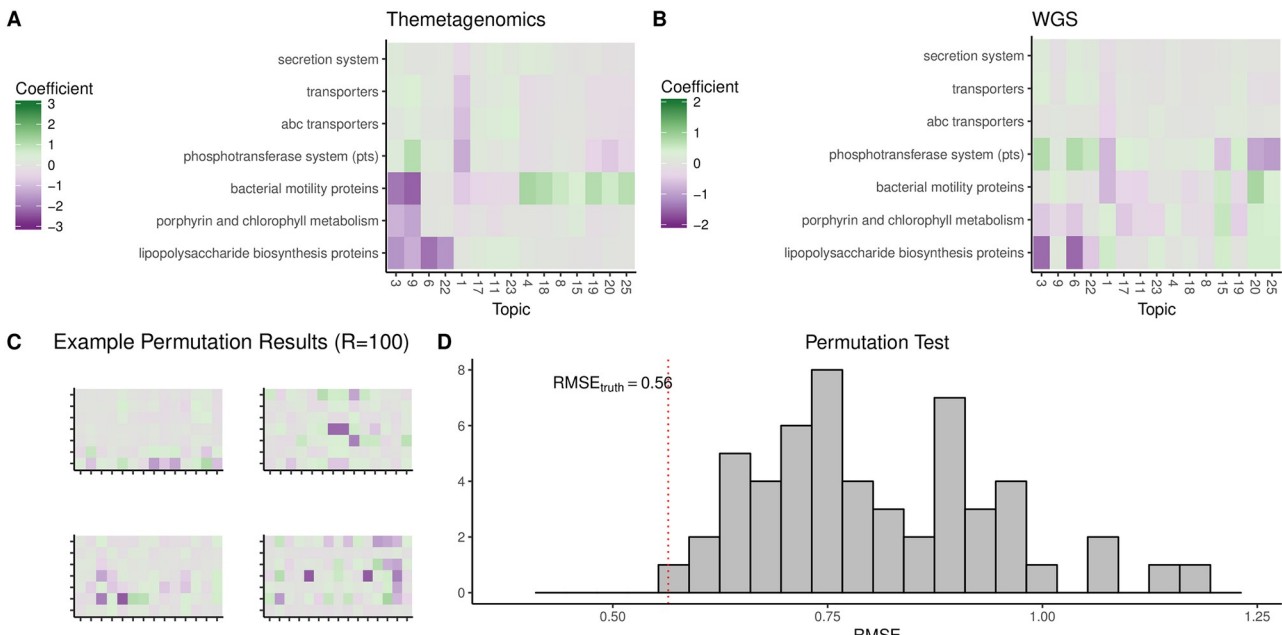

**Fig 4.** (A). KEGG (level-3) pathway category-topic interaction regression coefficients from the multilevel negative binomial model as a measure of association between pathway and topic. Only pathways present in both the themetagenomics analysis of 16S rRNA data and HUMAnN2 analysis of the metagenomics shotgun sequencing data are shown. Green = associated samples with positive cancer diagnosis, Purple = associated with healthy samples. (B). Pathway category-topic interaction regression coefficients for metagenomic data. Topics were generated based on KOs that belonged to high frequency taxa in the themetagenomics pipeline. (C). Example topic-pathway heatmaps, similar to Fig 4A and 4B from four of the 100 permuted metagenomic datasets using in the permutation test. (D). Distribution of root-mean-squared-error (RMSE) scores (between the topic-pathway interaction regression coefficients between themetagenomics and the metagenomic data) from the 100 permuted metagenomic datasets. The RMSE score (0.56) for the unpermuted metagenomic dataset is delineated by the red dotted line.

We performed a permutation test to determine whether the similarities in gene enrichments/depletions between themetagenomics and the metagenomic data were spurious. We randomly permuted the topic and gene pathway labels in the metagenomic data, refit the multilevel regression model, and then calculated the root mean square error (RMSE) for each topic-pathway interaction regression weight between the themetagenomics and permuted metagenomic models. After 100 replicate simulations, the RMSE for the unpermuted metagenomic model was smaller than every permuted metagenomic model ($p < 0.05$) (Fig 4C and 4D). Therefore, the apparent similarities in the gene enrichment/depletion profiles between themetagenomics and the shotgun data were not due to random chance, indicating that using predicted gene enrichment/depletion from 16S rRNA amplicon surveys resulted in similar within-topic predicted functional profiles to those obtained by directly measuring functional content via metagenomic shotgun sequencing.

### Detection of events in subject B from David et al. [23]

The David et al. [23] dataset contains fecal and salivary 16S rRNA gene surveys from two subjects. We focused on fecal samples from subject B. We compared our results to the three profiles described by David et al. [23], which consisted of a pre-food-poisoning profile (days 1–150), food-poisoning profile (151–159), and post-food-poisoning profile (150–318).

**The topic model approach identified 3 distinct gut configurations.** In the topic correlation network (Fig 5A), we identified a small subnetwork of three topics (marked by violet bracket) and two large subnetworks that contained 24 and 14 topics each (red and green brackets, respectively). The large subnetworks were connected by a chain of four topics (T9, T24, T2, T37) (blue bracket). We defined the four sets of correlated topics as topic clusters and sampled topic frequencies (across samples) and taxa frequencies (across topics) from the topic model's posterior distribution to assess how often topics and taxa occurred within these clusters.

Fig 5B shows the posterior frequency in which the topic clusters occurred given the day in which the sample was collected (the estimated posterior probability of a cluster occurring on a given day). There were two clear periods of rapid change in cluster frequency, specifically when transitioning from cluster 1 to 2 (days 152–154) and clusters 2 to 3 (day 161). Our intervals are similar to the original study's transition points at days 144–145 and 162–163, where the shift from a topic cluster 1 to topic cluster 2 corresponded with subject B's food poisoning diagnosis. The transition between topic clusters 1 and 2 is abrupt and likely occurred around day 153. Taxonomically, this transition is marked by a shift from Bacteroideaceae (posterior frequency = 0.338), Lachnospiraceara (0.276), and Rumunococcaceae (0.266) to Enterbacteriaceae (0.246) and Clostridiaceae (0.195) families (Fig 5D). In particular, day 153 was distinctive for topic 20. This rare topic was not correlated with any other topics and hence did not belong to any topic cluster. While its taxonomic profile was quite similar to topic cluster 1, it was distinctly enriched for *Enterobacteriaceaea spp.*, which is consistent with the subject's *Salmonella* diagnosis. Topic 20 likely marks the event of initial exposure to the pathogen.

The distribution of topic assignments for topic cluster 2 followed the order in which its topics were positioned in the topic correlation network (the linear chain of topics) (Fig 5E). The start of topic cluster 2, day 155, was dominated by topic 9, characterized by taxa substantially different from topic cluster 1. Bacteria enriched in this topic included *Haemophilus parainfluenzae*, *Clostridium perfringens*, and, notably, *Enterobacteriaceaea spp*. Thus, topic 9 likely represented the disrupted configuration of microbiota due to exposure to *Salmonella*. Enterbacteriaceae spp. and *C. perfringens*, via topic 24, continued to dominate on day 156. Day 157 was best described by topic 2, a topic rich in *Enterobacteriaceae spp*. as well as *Veillonella spp*. It should be noted, however, that our results were more conservative than David et al. [23] in

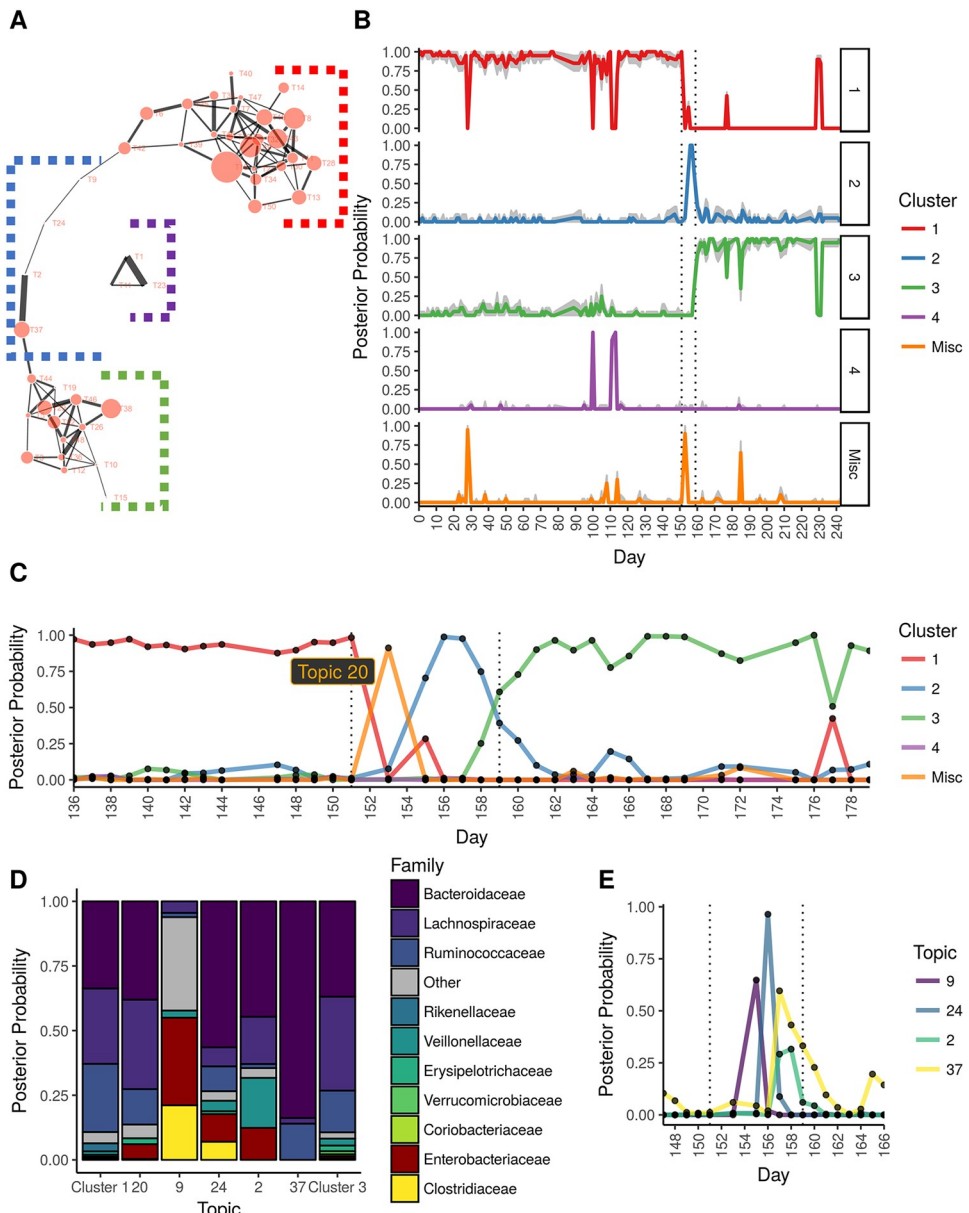

**Fig 5. Application of the topic model approach to David et al. [23] data.** (A). The topic-to-topic correlation graph showing two topic clusters (clusters 1 and 3) connected by a linear chain of topics (cluster 2) that follow the time progression of the taxonomic change due to the food poisoning infection. (B). Distribution of topic assignments as a function of day and cluster (panels), indicating 3 distinct profiles. The interval in which food poisoning symptoms presented (per David et al. [23]) are marked with dotted vertical lines. Gray shading indicated 80% uncertainty intervals. (C). Frequency of cluster assignments as a function of day, indicated day 153 marking the shift from profiles 1 to 2 and day 159 marking the shift from profiles 2 to 3. (D). Frequency of taxa assignments given a cluster assignment. Cluster 2 is shown in terms of its topics (9, 24, 2, 37). Topic 20 is also shown (misc. cluster), which lacked any edges in the correlation graph, but marks the initial appearance of *Enterobacteriaceae* on day 153 (representing the start of the infection). (E). The probability of the topic assignments given each day for cluster 2. The progression of topics also follows the progression of taxonomic change shown in the correlation graph.

that we confidently estimated that topic cluster 2 lasted roughly 4 days (155 to 158), which is much shorter than the original study's estimate (145 to 162). Our estimated length of illness (153 to 158) was more consistent to David et al. [23] (151 to 159), however. At approximately day 159, the taxonomic profile shifted toward cluster 3, which was similar to cluster 1 in terms

of Bacteroidaceae (0.369), but enriched in Lachnospiraceae (0.360) and depleted in Rumunoi-coccaceae (0.165) (Fig 5D).

**HC was unable to separate the transition between during- and post-illness periods.** We compared our approach to one using HC. HC cluster 4 contained 360 taxa and corresponded well to the pre-illness period, spanning days 1 to 150. The set of taxa was similar to the taxa identified in topic cluster 1 (Figure D in S1 File). The post-illness period was captured by HC clusters 1 and 3, but these clusters failed to completely separate the during- and post-illness periods; they spanned days 151 to 318.

## Limitations

There are limitations to our approach. First, the topic-pathway inference step currently scales poorly in terms of computation time for large numbers of topics, which may be more important as datasets grow. Regularization and sparsity-inducing priors help limit the number of important topics; hence, exploring only a subset of topics during the final regression step can offer substantial speed improvements at little cost, but utilizing the complete set of topic information would be ideal. Second, we are capable of separately estimating the uncertainty in our topic model, the multilevel regression model, and the functional predictions from PICRUSt, but we currently do not propagate the uncertainty throughout the pipeline. Doing so would improve downstream interpretation with better estimation of the uncertainty in topic-sample covariates and topic-pathway interactions, which in turn would greatly improve one's confidence in focusing on within-topic gene sets. Third, we do not incorporate phylogenetic branch length information, which could lead to more meaningful topics. Fourth, we can only provide our best estimations regarding how the model behaves with different sample and feature sizes. Our simulations (S1 Appendix) suggested that by effectively decreasing the number of samples, either through rarefying or using sparser data, power decreases. As expected, we were able to detect more sample effects using 500 samples compared to 100 samples. However, as we increased the sparsity of our feature space, our ability to capture known subcommunities was compromised, which would certainly have detrimental effects on detecting sample-level effects. The degree in which sparsity, feature space size, and sample size affect power and the ability of topics to capture meaningful co-occurrences will vary depending on the dataset, which in turns will vary in terms of diversity, range of counts, and sample-level information.

## Conclusion

We present our approach at a time when easily-to-interpret analyses for complex microbiome data are direly needed. Current methods often link the relative abundance of a single OTU to a sample information of interest (e.g., disease state). These methods routinely identify important subsets of taxa but ignore OTU co-occurrence and ratios. Network methods can overcome this concern, but typically don't incorporate phenotypic information within the model; consequently, they are incapable of directly linking sections of the OTU correlation network with sample metadata of interest. Constrained ordination methods, such as canonical correspondence analysis, do in fact couple inter-community distance with sample information, but the user is limited to specific distance metrics (e.g., Chi-squared) and must follow key assumptions (e.g., the distributions of taxa along environmental gradients are unimodal) [26]. Moreover, interpretation of biplots becomes increasingly difficult as more covariates are included. While linking key taxa to functional content can be accomplished via sparse canonical correlation analysis [27], this approach is susceptible to many of the interpretability problems found in other ordination approaches, and exploring inferred relationships in the context of taxonomic co-occurrence is not straightforward.

The ability to make meaningful inferences using current methods is further limited by the fact that microbiome data is often inadequately sampled (thus justifying some type of normalization procedure), compositional (due to normalization), sparse, and overdispersed. Thus, recent work has explored the use of Dirichlet-Multinomial models, which are well equipped at managing overdispersed count data [28–30]. The fact that Dirichlet-Multinomial conjugacy is exploited in many topics models hints at their applicability for relative abundance data. We selected the recently developed STM for our workflow because of its ability to not only utilize sample data as prior information as in the Dirichlet-Multinomial regression topic model [31], but also capture topic correlation structure and apply partial pooling over samples or regularization across regression weights.

Thus, we have proposed an approach for uncovering latent thematic structure in 16S rRNA amplicon data that provides a low-dimensional, biologically interpretable representation of taxonomic and predicted functional content. Rather than inferring functional content independently of taxonomic relative abundances, our approach shifts the focus to investigating within-topic functional content. Unlike other methods, by exploring our topics, we can link categories of functional content to specific clusters of taxa which can in turn be linked to sample features of interest. For example, like Gevers et al. [21], we detected a relationship between membrane transport genes and CD+, but our approach also allowed us to determine which bacteria (OTUs belonging to Enterobacteriaceae) were the prime contributors to the enrichment of membrane transport genes. Moreover, the pathogenic set of bacteria reported by Gevers et al. [21] (*Haemophilus* spp., *Neisseria*, and *Fusobacteria*) contributed less to the predicted abundance of membrane transport genes. By independently applying statistical approaches to the OTU and predicted functional content, as is typical, the apparent relationship between membrane transport genes and specific configurations of bacteria would be lost.

We have also shown that our approach drastically reduces the dimensionality of two high-dimensional sources of information, taxonomic relative abundances and predicted functional content, increasing the ease in which these data can be interpreted. For Gevers et al. [21], we determined that T15 is (1) associated with CD+ samples; (2) dominated by a cluster of bacteria previously associated with CD; and (3) uniquely enriched for a subset of LPS synthesis genes. With a gene profile from a topic of interest, one could focus on gene subsets associated with topic-specific bacterial clusters that are known disease biomarkers, which in turn may facilitate targeted approaches for future research endeavors.

Lastly, our complete pipeline is computationally manageable. Fitting the topic model to a dataset with nearly 5000 samples reached convergence in minutes. Functional prediction via PICRUSt also only takes minutes (using our C++ implementation in themetagenomics). Inferring topic-pathway interactions via our multilevel, negative binomial regression approach is comparatively slower, however, taking hours for large datasets. However, this is still manageable. Thus, we offer a viable package that can help researchers discover configurations of taxa and functions that correlate to sample metadata. This is because we implement this model in the probabilistic programming language Stan, which uses Hamiltonian Monte Carlo. Maximum likelihood (a much faster alternative) does not provide estimates of uncertainty and generally fails to converge for these data, although the regression weight estimates tend to be quite similar based on our experience.

## Methods

### Review of the Structural topic model

The STM [20] is a Bayesian generative topic model. It begins with a given a set of M samples, each consisting of N OTUs. These N OTUs are, in turn, elements of a fixed vocabulary of V

unique OTU IDs. From this, K (a fixed number chosen a priori) latent topics are assumed to be generated from the data. These topics consist of overlapping sets of co-occurring OTUs. Note that we will describe the STM in the context of the analyses perform herein; for a complete description of the STM, see [20]. The observations include the presence of OTU $w_n$ occurring in sample m and an $M \times P$ matrix of sample-level information such as disease state or age.

For our purposes, the posterior distribution of unobserved (latent) parameters given the observed data is given by:

$$\text{Posterior Distribution}: p(\theta, \beta, \Sigma, \Gamma, zw, X).$$

The generative process is formulated by first specifying the probability

$$P(\text{Topic } k \text{ occurs in Sample } m) = \theta_{m,k}, \sum_{k=1}^{K} \theta_{m,k} = 1$$

and, for each of the samples, is assumed to follow logistic normal distributions,

$$\theta \sim LN_{K-1}(\Gamma^T X_m^T, \Sigma)$$

where $\Gamma$ is a $P \times (K-1)$ matrix of regression coefficients that estimate the degree of influence a covariate $X_p$ has on $\theta$; and $\Sigma$ is a $K \times K$ covariance matrix. In addition to $\theta$, the probability

$$P(\text{OTU } n \text{ occurs in Topic } k) = \beta_{k,n}, \sum_{n=1}^{N} \beta_{k,n} = 1$$

For each topic, $\beta_k$ is assumed to be Dirichlet distributed. Finally, both topic assignments $z_{m,n}$ for each OTU $w_{m,n}$, along with each OTU, obey multinomial distributions,

$$z_{m,n} \sim \text{Multinomial}(\theta_m)$$

$$w_{m,n} \sim \text{Multinomial}(\beta, z_{m,n})$$

For the relationships between topic model nomenclature and our terminology, see Table 1. The posterior distribution is estimated by a semi-collapsed variational expectation maximization procedure. Convergence is reached when the relative change in the variational objective (i.e., the estimated lower bound) in successive iterations falls below a predetermined tolerance.

## Empirical datasets

The Gevers et al. [21] dataset (PRJNA237362, 03/30/2016) is a multicohort, IBD dataset that includes 16S rRNA amplicon data from control, CD, and ulcerative colitis samples taken from multiple locations throughout the gastrointestinal tract. The Schmidt et al. [22] dataset (PRJEB4953, 08/14/2017) consists of human oral microbiota obtained from control subjects and subjects diagnosed with oral cancer. These samples underwent 16S rRNA amplicon sequencing, and a subset (N = 12) also underwent metagenomic shotgun sequencing.

## 16S rRNA amplicon data preparation and OTU picking

Paired-end reads were joined and quality filtered via QIIME v 1.9.1 and dada2 for Gevers et al. [21] and Schmidt et al. [22] data, respectively. Closed-reference OTU picking was performed with QIIME using SortMeRNA against GreenGenes v13.5 at 97% sequence identity. This was followed by copy number normalization via PICRUSt version 1.0.0 [32]. Samples with fewer than 1000 total reads were omitted. OTUs that lacked a known classification at the phylum level were removed. For Gevers et al. [21], we selected only terminal ileum samples and filtered

OTUs with fewer than 10 total reads across samples, yielding 555 samples over 1500 OTUs. For Schmidt et al. [22], we filtered any OTU with non-zero abundances in fewer than two samples, yielding 81 samples over 1029 OTUs.

## Metagenomic shotgun sequence data preparation and functional genomic profiling

Low quality reads and human genomic sequences were filtered via KneadData. Functional profiles were then generated using HUMAnN2 with the ChocoPhlAn nucleotide database and UniRef90 protein database. The UniRef90 protein families were collapsed into KEGG orthologies (KOs), yielding abundances (copies per million (CPM)) for 12 samples over 36,806 KOs.

## Structural topic model fitting

The OTU abundance tables consisted of counts normalized by 16S rRNA gene copy number (to be consistent with the PICRUSt approach for functional prediction, which we use downstream after model fitting). No other normalization (e.g., rarefying, DESeq2 method, or dividing by total reads) was performed based on the simulation results in (S1 Appendix). STMs with different parameterizations in terms of topic number (K ∈ 15, 25, 50, 75, 100, 150, 250) and sample features (e.g., no features, indicators for presence of disease, diet type, etc.) were fit to the OTU tables generated from Gevers et al. [21] data via the R package stm [33]. We evaluated each model fit for presence of overdispersed residuals and conducted permutation tests (permTest in the stm package) where the sample feature of interest is randomly assigned to a sample prior to fitting the STM. To compare parameterizations between models, we evaluated predictive performance using held-out likelihood estimation [34].

## Assessing topic generalizability

We performed classification to assess the generalizability of the extracted topics. No sample information was used as covariates in the logistic normal component of the STM. Samples were split into 80/20 training-testing datasets. For different number of topics (K ∈ 15, 25, 50, 75, 100, 150), an STM was trained to estimate the topics-over-OTUs distribution (β). We then held this distribution fixed; hence, only the testing set's samples-over-topics distribution (θ) was estimated. For both the training and testing sets, simulated posterior samples from the samples-over-topics distribution (θ) were averaged. The resulting posterior topic frequencies in the training set were then used as features to classify sample labels, similar to using $\bar{Z}$ in supervised LDA [35]. Generalization (testing) error was assessed using the optimal parametrization based on cross-validation performance on the test set topic frequencies. Classification was performed using a random forest classifier, which underwent parameter tuning to determine the number of variables for each split. This was accomplished through repeated (10x) 10-fold cross-validation, using up-sampling to overcome class imbalance. We performed a parameter sweep over the number of randomly selected OTU features, while setting the number of trees fixed at 128. The optimal parameterizations were selected based on maximizing ROC area under the curve.

 The performance of the STMs was compared to the performance using OTUs as features from the original OTU abundance table. Separately, training and testing set OTU abundances were converted to relative abundances with the following equation: $OTU_{n,m}/\Sigma_n OTU_{n,m}$. In words, OTU $n$ for sample $m$ is scaled by the library size of sample $m$ (the total abundance of sample $m$). The resulting OTU relative abundance tables were separately z-score normalized. Training cross-validation and testing using a random forest was then performed as above.

## Identifying within-topic clusters of high frequency OTUs

Using the topics-over-OTUs distribution, we performed hierarchical clustering via Ward's method on Bray-Curtis distances. We refer to high frequency groups of OTUs as "clusters."

## Inferring within-topic functional potential

We obtained the topics-over-OTUs distribution (β) for each fitted model and mapped the within-topic OTU probabilities to integers ("pseudo-counts") using a constant: $10000 \times \beta$. A large constant was chosen to prevent low frequency OTUs from being set to zero, although their contribution to downstream analysis was likely negligible. Gene prediction was performed on each topic-OTU pseudo-count table using PICRUSt version 1.0.0 [14]. (Normalization of 16S copy number was performed prior to topic model fitting using PICRUSt.) Predicted gene content was classified in terms of KOs [36].

## Identifying topics of interest

Topics of interest were identified using the samples-over-topics distribution, where each column represents the frequency of topic $k$ for each sample. Each column was regressed against CD diagnosis. We calculated 95% uncertainty intervals using an approximation that accounts for uncertainty in estimation of both the sample covariate coefficients and the topic frequencies. We refer to these coefficients as "topic-sample-effects." Coefficients whose 95% uncertainty intervals do not span 0 are referred to as "high-ranking-topics."

## Validating within-topic co-occurrence

To determine how well the high-ranking-topics captured co-occurrence in the original OTU relative abundance table, we sampled the top-10 highest frequency taxa in each high-ranking topic's topics-over-OTUs distribution (β). We then normalized the original OTU table using the centered-log-ratio transformation and then evaluated how the high frequency taxa vary as a function of CD diagnosis and PCDAI.

## Posterior inference

To determine how well the high-ranking-topics captured the taxonomic profile associated with CD, we performed the following posterior simulation over R = 1000 iterations. First, for iteration r, for all samples $m \in M$ (e.g., subject 134), we obtained 100 posterior samples ($i \in \{1, \ldots, 100\}$) of $\theta_m^{(i)}$ from the posterior distribution, $p(\theta, \beta, \Sigma, \Gamma, z | w, X)$. For each of these $\theta_m^{(i)}$, we sampled topic assignments $z_{m,n}^{(i)} \sim \text{Multinomial}(\theta_m^{(i)})$, and then OTUs $\hat{w}_{m,n}^{(i)} | z_{m,n}^{(i)} \sim \text{Multinomial}(z_{m,n}^{(i)}, \beta)$.

We then recorded whether the topic assignments $z_{m,n}^{(i)}$ belonged to one of the high-ranking-topics and whether they have a positive or negative association with sample covariates of interest, resulting in positive-, negative-, and no-association topic groups. We calculated the frequency $f_n^{(g)}$ in which OTUs $\hat{w}_{m,n}^{(i)}$ were sampled from a given topic group g:

$$f_n^{(g)} = \sum_i \sum_{\hat{w}_{m,n}^{(i)} | z_{m,n}^{(i)}} 1[z_{m,n} \in g]$$

where $1[\cdot]$ is the indicator function. For each OTU, we calculated which group had the largest

sampling frequency:

$$f_n^{(g)*} = 1 \left[ f_n^{(g)} = \underset{g}{\mathrm{argmin}} \ f_n \right]$$

After 1000 iterations, we calculated

$$F_n^{(g)*} = \frac{1}{R} \sum_r f_n^{(g)*(r)}$$

For each topic group, we extracted a subset of OTUs that had frequencies above 0.99. In the original relative abundance table, for each sample, we calculated the relative abundance of each group of OTUs.

## Identifying functional content that distinguishes topics

To determine which predicted functional gene content best distinguished topics, we used the following multilevel negative binomial regression model:

$$\theta_{k,c} = \exp[\mu + \beta_k + \beta_c + \beta_{k,c}]$$

$$y_{k,c} \sim \mathrm{NB}(\theta_{k,c}, \lambda)$$

where $\mu$ is the intercept, $\beta_k$ is the per topic weight, $\beta_c$ is the per level-3 gene category weight, $\beta_{k,c}$ is the interaction weight for a given topic-function (gene category) combination, $y_{k,c}$ is the count for a given topic-function combination, and $\lambda$ is the dispersion parameter. The intercept $\mu$ was given a Normal(0, 10) prior; all weights were given Normal(0, 2.5) priors; and the dispersion parameter $\lambda$ was given a Cauchy(0, 5) prior. Model inference was performed using Hamiltonian Monte Carlo in the R package rstanarm [37]. Convergence was evaluated across four parallel chains using diagnostic plots to assess mixing and by evaluating the Gelman-Rubin convergence diagnostic [38]. To reduce model size, we used genes belonging to only 15 (arbitrary number) level-2 KEGG pathway categories (Table E in S1 File). For large topic models, we fit only the top 25 topics, ranked in terms of topic-sample-effects that measure the degree of association between samples-over-topics probabilities and our sample feature of interest.

## Assessing relationships between sample information of interest and taxonomic relative abundance

To quantify the relationship between taxonomic relative abundance and continuous sample features (such as PCDAI), we performed negative binomial regression (log-link), using sample library size (sum of OTU abundances across samples) as an offset. The family-wise error rate was adjusted via Bonferroni correction. Significance levels for hypothesis testing was set at 0.05.

## Comparing within-topic functional profiles to an OTU-relative-abundance-based approach

We compared the results from the hierarchical negative binomial model to a differential abundance approach. We performed predicted functional content using PICRUSt on copy number normalized OTU abundances. The resulting functional abundances were collapsed into level-3 KEGG pathways. Note that, for consistency, we again restricted the KOs to the 15 level-2 KEGG pathways used previously. The resulting level-3 pathway abundances underwent

DESeq2 differential abundance analysis, which uses negative binomial regression and variance stabilizing transformations to infer the difference log-fold change of OTU relative abundance [7,8]. The resulting p-values were corrected via the Bonferroni method. Adjusted p-values below 0.1 were considered significant.

### Fitting BioMico

The same training and testing sets were used as described above. Assemblages of 25 and 50 were trained with default parameters unless specified: burnin = 5000, delay = 500 (25 assemblages) or delay = 100 (50 assemblages), rarefaction_depth = 1000. Parameters were adjusted to decrease training time to less than 3 days. Posterior distributions were evaluated to ensure MCMC convergence.

### Validating extracted functional profiles using metagenomic shotgun sequencing data

The themetagenomics pipeline was applied to the Schmidt et al. [22] OTU table: (1) data were normalized for 16S rRNA gene copy number; (2) normalized OTU abundances were fit using a 25 topic STM with cancer diagnosis as a binary covariate; (3) within-topic functional content was predicted using PICRUSt; and then (4) topic-pathway effects were detected using the multilevel regression model.

For each topic, we identified the high probability OTUs (those with frequencies greater than 1% in that topic), obtained their genus classification, and then subset the metagenomic KO table such that only KOs corresponding to these genera are present. Then, for each level-3 KEGG pathway, we summed the abundances of all remaining KO members. Topic-pathway effects were then detected with the following multilevel regression model:

$$\theta_{k,c} = \exp[\mu + \beta_1 X + \beta_k + \beta_c + \beta_{k,c} + \log Z]$$

$$y_{k,c} \sim \mathrm{NB}(\theta_{k,c}, \lambda)$$

where X is a binary column vector indicating positive cancer diagnosis, $\beta_1$ is the coefficient for cancer diagnosis, and log $Z$ is an offset accounting for sample library size (sample sum). The remaining parameters are analogous to the model described above.

A permutation test was performed to compare the similarity in topic-pathway effects between themetagenomics and the metagenomic model to random sampling. In the metagenomic KO table, topic and pathway labels were randomly permuted. The permuted table was then refit with the regression model described. The root mean squared error was calculated between the topic-pathway regression coefficient $\beta_{k,c}$ for themetagenomics and the metagenomic model:

$$RMSE = \sqrt{\frac{\sum_{k,c} \left(\beta_{k,c}^{(theme)} - \beta_{k,c}^{(\mathrm{meta})}\right)^2}{n}}$$

This process was repeated over 100 permuted replicates to calculate a null distribution of RMSE scores, which was then compared to the true RMSE between the unpermuted metagenomic KO table and *themetagenomics*. A p-value ($\alpha = 0.05$) was calculated as the proportion of RMSE scores from the 100 permuted metagenomic KO tables that were less than the RMSE score for the unpermuted metagenomic KO table.

### Exploring thematic structure in David et al. [23]

**Data preparation and OTU picking.** The David et al. [23] dataset contains fecal and salivary 16S rRNA surveys from two subjects. The samples were obtained at uneven sampled times from 318 days. Data from were downloaded from the European Bioinformatics Institute (EBI) European Nucleotide Archive (ENA) (accession number ERP006059). It consisted of 1.7 million 16S rRNA gene (V4 region) sequencing reads, 100 bp in length. The reads were quality filtered using the fastqFilter command in the dada2 package [39]. Closed reference OTU picking was then performed with QIIME version 1.9.1. using SortMeRNA again GreenGenes v13.5 at 97% sequence identity [24].

**Data preprocessing and STM fitting.** From the OTU table, we removed any samples with fewer than 1000 total reads, were not of fecal origin, were not from donor B, and did not include sample data for day, donor, and body site. OTUs lacking a known phylum classification or present in fewer than 1% of the remaining samples were removed. The remaining OTUs were normalized in terms of 16S rRNA gene copy number per the table provided by PICRUSt [14]. The final OTU table consisted of 1562 OTUs across 189 samples.

We fit 7 STMs that varied in terms of topic number $K \in \{15, 25, 50, 75, 105, 155, 250\}$. To infer the relationship between sample data and the samples-over-topics distribution θ, we used two sample covariates: two continuous, integer valued sequences representing the day number in the sequence and the DOW. Given our assumption that fluctuations in microbiota likely varied nonlinearly with respect to day, we used a smoothing spline with 10 degrees of freedom on day and a second-degree polynomial on DOW.

**Event detection.** To detect events in subject B, we repeated the approach described for simulation 2 (S1 Appendix).

**Hierarchical clustering.** We performed HC for comparison. The David et al. [23] data were normalized using the sample geometric mean to correct for library size imbalance. Each feature was then centered and scaled as described for simulation 2. Clustering was performed as detailed for simulation 2. The resulting tree was cut to produce 6 clusters. The choice of 6 clusters was based on the three profiles identified by David et al. [23] (days 1–150, 151–159, and 160–318). We included three additional clusters to account for the background taxonomic variation lacking one of the three profiles of interest. Because we are basing our parameter choice on what can be considered the truth, this can be considered a best-case-scenario.

**Themetagenomics.** An R package, where the user provides a taxonomic abundance table, sample-level information, and taxonomy information. A topic model is fit, and the user is provided interactive Shiny applications, allowing the user to visualize, graphically, (1) which taxa dominant which topics, and hence which taxa co-occur; (2) which functional content dominant which topics, and (3) relationships between sample-level information (e.g., age) and the relative frequency of co-occurring taxa. The figures can be exported as image files.

## Supporting information

**S1 Appendix. Contains additional information regarding the following: (1) simulation 1 which explores different normalization approaches, (2) time series analysis methods for David et al. [23] data including simulation 2; and (3) additional results for Crohn's disease data as well as expansion of results detailed above and comparisons to other approaches such as SPIEC-EASI.**
(DOCX)

**S1 File. Contains figures A-D and tables A-F.**
(DOCX)

## Author Contributions

**Conceptualization:** Stephen Woloszynek, Joshua Chang Mell.

**Data curation:** Stephen Woloszynek, Zhengqiao Zhao.

**Formal analysis:** Stephen Woloszynek, Zhengqiao Zhao.

**Funding acquisition:** Gail L. Rosen.

**Investigation:** Stephen Woloszynek, Zhengqiao Zhao, Gail L. Rosen.

**Methodology:** Stephen Woloszynek, Joshua Chang Mell, Zhengqiao Zhao, Gideon Simpson, Michael P. O'Connor, Gail L. Rosen.

**Resources:** Gail L. Rosen.

**Software:** Stephen Woloszynek.

**Supervision:** Stephen Woloszynek, Joshua Chang Mell, Gideon Simpson, Michael P. O'Connor, Gail L. Rosen.

**Validation:** Stephen Woloszynek, Zhengqiao Zhao.

**Visualization:** Stephen Woloszynek, Zhengqiao Zhao.

**Writing – original draft:** Stephen Woloszynek, Zhengqiao Zhao.

**Writing – review & editing:** Stephen Woloszynek, Joshua Chang Mell, Zhengqiao Zhao, Gideon Simpson, Michael P. O'Connor, Gail L. Rosen.

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
