## [Decision Letter · Decision Letter 0]

12 Aug 2019

PONE-D-19-17255

Themetagenomics: Exploring Thematic Structure and Predicted Functionality of 16s rRNA Amplicon Data

PLOS ONE

Dear Mr. Woloszynek,

Thank you for submitting your manuscript to PLOS ONE. After careful consideration, we feel that it has merit but does not fully meet PLOS ONE’s publication criteria as it currently stands. Therefore, we invite you to submit a revised version of the manuscript that addresses the points raised during the review process.

We would appreciate receiving your revised manuscript by Sep 26 2019 11:59PM. To enhance the reproducibility of your results, we recommend that if applicable you deposit your laboratory protocols in protocols.io, where a protocol can be assigned its own identifier (DOI) such that it can be cited independently in the future. For instructions see: http://journals.plos.org/plosone/s/submission-guidelines#loc-laboratory-protocols

We look forward to receiving your revised manuscript.

Kind regards,

Juan J Loor

Academic Editor

PLOS ONE

Reviewers' comments:

Reviewer's Responses to Questions

**Comments to the Author**

1. Is the manuscript technically sound, and do the data support the conclusions?

Reviewer #1: Yes

Reviewer #2: Yes

Reviewer #3: Partly

2. Has the statistical analysis been performed appropriately and rigorously? 

Reviewer #1: Yes

Reviewer #2: Yes

Reviewer #3: Yes

3. Have the authors made all data underlying the findings in their manuscript fully available?

Reviewer #1: Yes

Reviewer #2: Yes

Reviewer #3: Yes

4. Is the manuscript presented in an intelligible fashion and written in standard English?

Reviewer #1: Yes

Reviewer #2: Yes

Reviewer #3: Yes

5. Review Comments to the Author

Reviewer #1: The manuscript by Woloszynek et al. presented an interesting tool based on STM to link co-occurring groups of taxa with sample metadata and enable to analyze the interactions between topics and pathways. The tool is a good addition to or extension of several recently developed balance-based approaches, such as selbal (Rivera-Pinto et al., 2018).

1. When the discussion of limitations of the approach is particularly interesting, the authors may consider adding the discussion of the minimal value for M (samples) and/or sequencing depth L560 and its implications.

2. For the tool to be widely utilized by bench biologists in the microbiome community, the reviewer strongly recommended to publish a Galaxy version.

3. It would be very helpful if the authors add a section on typical output formats/files so that readers can have a better evaluation of the tool utility without actually running the algorithm.

4. The provided link https://github.com/EESI/themetagenomics has not been updated and it does not include a user-friendly manual.

Reviewer #2: Comments to the Author

Overview

This paper describes a workflow and a tool to conduct a topic model based inference of microbial taxonomic and functional signatures of specific metadata from 16S rRNA gene amplicon sequencing data. Although shotgun metagenomics become cheaper and more common, still the amplicon sequencing is widely used in microbial community analyses. The topic model is already applied to the taxonomic composition data of microbial community in some studies [Chen et al. 2011, Shafiei et al. 2015, Yan et al. 2017, Higashi et al. 2018]. But this manuscript and the R-based package "themetagenomics" provide seamless workflow of taxonomic composition data to taxonomic and functional signatures of specific metadata. This package will promote the topic model -based microbial community analyses for many researchers.

General Comments

In this manuscript, the reviewer thinks different taxonomic ranks (e.g, species, genus, family) are mixed without any consideration. The taxonomic assignment threshold of 16S rRNA gene sequences in this manuscript is based on the 97% sequence identity. This threshold is usually for species assignment. Therefore, the taxonomic composition should be the species composition (although nearly half of sequences generally cannot be assigned to species).

However, the combination of QIIME and GreenGenes produce the problematic mixed taxonomic rank composition.

In the taxonomic composition data in this manuscript, which types of taxonomic composition are used for topic model analyses?

(i) the abundance of each taxon (e.g., Roseburia) is the sum of the abundances of more detailed taxa (e.g., Roseburia intestinalis, R. hominis, etc.) and the abundance of that taxon (e.g., Roseburia).

(i.e., Roseburia abundance = R. intestinalis abundance (150 sequences) + Roseburia abundance (50 sequences)).

(ii) the abundance of each taxon (e.g., Roseburia) is independent of the abundances of more detailed taxa (e.g., Roseburia intestinalis, R. hominis, etc.).

(i.e., Roseburia abundance = Roseburia abundance (50 sequences)).

Both types of composition can be calculated from the QIIME and GreenGenes based taxonomic assignment results.

Line 397

GreenGenes taxonomic hierarchy (GreenGenes), ChocoPhlAn taxonomic hierarchy (NCBI Taxonomy), and KEGG taxonomic hierarchy (KEGG) are different.

For example, the species belong to a genus Clostridium are different among three taxonomic hierarchies.

How to identified all reads belonging to the high frequency taxa (taxonomic name is based on GreenGenes) of each topic in the metagenomic shotgun data (taxonomic name is based on NCBI Taxonomy or KEGG)?

Specific Comments

Title

16s  16S

Line 70

The authors describe "the normalization transforms the abundances into relative abundances within each sample (compositional data)". However, the sequenced reads data are generally already relative abundance data.

When the sequencing, almost all of sequencing run contains multiple samples. Then, to avoid extremely different read number per sample, the DNA molecular weight per sample is normalized among samples within each sequencing run by wet experiments. Therefore, the number of sequenced reads generally does not comparable among samples directly without the normalization of total number of reads or the subsampling of reads.

Line 477

Supplementary Figure number is wrong.

Line 614

Normalization of number of total reads per sample or subsampling of reads was not performed?

Line 681

Mulinomial  Multinomial

Reviewer #3: 1. The “Thematogenomics” terminology seems rather an arbitrary measurement of the co-occurring microbial functional attributes in a microbial community without a strong scientific basis. The authors should revisit the title and consider revising. It is a “catchy” title but lacks a strong scientific basis for which the authors attempted to establish a unique bioinformatics tool.

2. The analysis is based upon the HTS of 16S rRNA gene amplicons within the microbial metacommunity DNA to determine predictive metabolic function like the PiCrust analysis of a co-occurring subset of the microbial community (based on the CoNet analysis). The CoNet analysis may produce highly variable keystone taxa including the nodes on the same HTS sample. Thus the reproducibility of the CoNet results may not be achieved.

3. How do the microbial functional attributes and co-occurring microbial community from 16S rRNA amplicon HTS data if conducted compares to the shotgun metagenomics? Would microbial taxa (OTUs) be different between the two approaches? Validation is necessary.

4. The primary objective of this study seems the development of a new and unique bioinformatics tool to precisely identify the co-occurring microbial sub-community and the keystone taxa and then use them to determine the predictive functional attributes. The key issue with this manuscript is that the approach uses bias when pre-selecting the functions. If that is correct then how this approach would address the outcome of unknown sample properties during double-blinded studies? There seem to be a flaw in this approach, particularly when unknown samples are tested. The authors should address this limitation.

5. The study needs to use more than one sample type (multiple clinical as well as environmental) in order to establish a broadly applicable bioinformatics tool. Moreover, the authors used the HTS files available on the open-access database. However, for this study, the authors may consider designing experiments with variable sample parameters to demonstate the modulation of the co-occurring microbial community as well as the keystone taxa resulting into he changes in microbial metabolic functional attributes.

6. PLOS authors have the option to publish the peer review history of their article (what does this mean?). If published, this will include your full peer review and any attached files.

Reviewer #1: No

Reviewer #2: No

Reviewer #3: No

---

## [Author Response · Author response to Decision Letter 0]

25 Sep 2019

See attached document for color coded version.

Reviewer #1: The manuscript by Woloszynek et al. presented an interesting tool based on STM to link co-occurring groups of taxa with sample metadata and enable to analyze the interactions between topics and pathways. The tool is a good addition to or extension of several recently developed balance-based approaches, such as selbal (Rivera-Pinto et al., 2018).

1. When the discussion of limitations of the approach is particularly interesting, the authors may consider adding the discussion of the minimal value for M (samples) and/or sequencing depth L560 and its implications. -- We added a 4th limitation to the limitations section to address this. We mentioned some of our simulation results from S2_appendix to address this and expanded on that as well. See lines 494-504 of the manuscript.

2. For the tool to be widely utilized by bench biologists in the microbiome community, the reviewer strongly recommended to publish a Galaxy version. -- We would love to have our package available in other languages and software. We made our selection of R carefully: (1) the STM, which we argue is the topic model of choice for our pipeline, already has a very well written package available in R which we could easily use; (2) R is a popular language for both computational and bench biologists since it has bioconductor and phyloseq, among other packages; (3) R is open source, free to use, and easy to write basic scripts in. We have our package easily available on CRAN -- it and its dependencies are easily installed in R. With that said, we are open to a galaxy implementation of our package -- as it seems straightforward to run R packages in Galaxy. We currently are working on releasing a 1.0 version of themetagenomics in R given the feedback we have received. Once that is complete, we will focus on other ports.

3. It would be very helpful if the authors add a section on typical output formats/files so that readers can have a better evaluation of the tool utility without actually running the algorithm. -- We added a section at the end of the methods section named “themetagenomics” that describes the input files needed and the outputs generated.

4. The provided link https://github.com/EESI/themetagenomics has not been updated and it does not include a user-friendly manual. -- Themetagenomics is an R package that will have future updates. That site houses the currently working, publicly available code. Because it’s an R package, the package is housed on CRAN where all packages are available (https://cran.r-project.org/web/packages/themetagenomics/index.html) and that page contains basic information as well as vignettes for the user to learn about the package. We have added the CRAN link to the manuscript.

Reviewer #2: Comments to the Author

Overview

This paper describes a workflow and a tool to conduct a topic model based inference of microbial taxonomic and functional signatures of specific metadata from 16S rRNA gene amplicon sequencing data. Although shotgun metagenomics become cheaper and more common, still the amplicon sequencing is widely used in microbial community analyses. The topic model is already applied to the taxonomic composition data of microbial community in some studies [Chen et al. 2011, Shafiei et al. 2015, Yan et al. 2017, Higashi et al. 2018]. But this manuscript and the R-based package "themetagenomics" provide seamless workflow of taxonomic composition data to taxonomic and functional signatures of specific metadata. This package will promote the topic model -based microbial community analyses for many researchers.

General Comments

In this manuscript, the reviewer thinks different taxonomic ranks (e.g, species, genus, family) are mixed without any consideration. The taxonomic assignment threshold of 16S rRNA gene sequences in this manuscript is based on the 97% sequence identity. This threshold is usually for species assignment. Therefore, the taxonomic composition should be the species composition (although nearly half of sequences generally cannot be assigned to species).

However, the combination of QIIME and GreenGenes produce the problematic mixed taxonomic rank composition.

In the taxonomic composition data in this manuscript, which types of taxonomic composition are used for topic model analyses?

(i) the abundance of each taxon (e.g., Roseburia) is the sum of the abundances of more detailed taxa (e.g., Roseburia intestinalis, R. hominis, etc.) and the abundance of that taxon (e.g., Roseburia).

(i.e., Roseburia abundance = R. intestinalis abundance (150 sequences) + Roseburia abundance (50 sequences)).

(ii) the abundance of each taxon (e.g., Roseburia) is independent of the abundances of more detailed taxa (e.g., Roseburia intestinalis, R. hominis, etc.).

(i.e., Roseburia abundance = Roseburia abundance (50 sequences)).

Both types of composition can be calculated from the QIIME and GreenGenes based taxonomic assignment results. -- We use an OTU table, generated via QIIME with GreenGenes at 97% sequence identify, which is loosely considered a “species cutoff,” but just as often includes 16S rRNA from many species or even more than one genus. Once those OTUs and their abundances are put into the table (which is what themetagenomics operates on), taxonomic labels are assigned to the OTUs (via Greengenes in QIIME). For many OTUs, no species-level assignment exists and it’s common not to have OTUs that have genus- or even order- level labels. Only a subset of taxonomies in GreenGenes (or other databases) even report at the species level. But most importantly, we are not fitting the model at any particular taxonomic level. Instead, OTU abundances are used. The OTUs can be mapped to their taxonomic annotations using the Greengenes taxonomic hierarchy provided by QIIME. Doing this mapping does not affect the model fit, since that has already been done. Because we know which OTUs belong to a given topic and their frequency (how frequent they are in that topic), and we know each OTU’s taxonomic hierarchy (again, via QIIME, typically with high-confidence assignments to the genus level), we can extrapolate the frequency of any given family (or other taxonomic level) in that topic as well. 

Line 397

GreenGenes taxonomic hierarchy (GreenGenes), ChocoPhlAn taxonomic hierarchy (NCBI Taxonomy), and KEGG taxonomic hierarchy (KEGG) are different.

For example, the species belong to a genus Clostridium are different among three taxonomic hierarchies.

How to identified all reads belonging to the high frequency taxa (taxonomic name is based on GreenGenes) of each topic in the metagenomic shotgun data (taxonomic name is based on NCBI Taxonomy or KEGG)? -- It is true that there are numerous different extant taxonomic hierarchies, and indeed they are under continued revision. We use GreenGenes, because it has the most comprehensive database of 16S rRNA gene sequences. Fortunately, we made our cross-data comparisons in the functional space, and therefore, we did not have to rely on the differing taxonomies in our comparison of 16S rRNA data to metagenomic data. We converted the 16S rRNA to the functional capacity of its genome via PICRUSt and compared to the functional capacity of the metagenome via KEGG Orthology. (While PICRUSt software converts 16S rRNA to KEGG orthologies, the metagenome data was mapped to UniRef90 protein families that mapped into KEGG orthologies (KOs)). So, all comparisons were on the functional level. Mapping between taxonomies is beyond the scope of this manuscript, though we note that the DSMZ maintains an up-to-date list that includes synonyms (https://www.dsmz.de/services/online-tools/prokaryotic-nomenclature-up-to-date), and some studies have attempted to map different taxonomies onto each other (https://bmcgenomics.biomedcentral.com/articles/10.1186/s12864-017-3501-4). .

Specific Comments

Title

16s  16S -- done

Line 70

The authors describe "the normalization transforms the abundances into relative abundances within each sample (compositional data)". However, the sequenced reads data are generally already relative abundance data.

When the sequencing, almost all of sequencing run contains multiple samples. Then, to avoid extremely different read number per sample, the DNA molecular weight per sample is normalized among samples within each sequencing run by wet experiments. Therefore, the number of sequenced reads generally does not comparable among samples directly without the normalization of total number of reads or the subsampling of reads. -- Yes. We are not challenging normalization and its importance. We are saying that methods that simply subsample or normalize a sample’s reads by the total number of reads in a sample present their own issues (and that is why normalization method should be carefully chosen), which are detailed in the cited papers, notably 5-8, which are all cited in this section.

Line 477

Supplementary Figure number is wrong. -- figure number is correct. It is figure S4 in the supplementary figures file, so “Figure S4. Clusters via hierarchical clustering (k=12) applied to the David et al. dataset (subset B). Red lines signify the presentation of illness” in “S1_supporting_figures.docx”

Line 614

Normalization of number of total reads per sample or subsampling of reads was not performed? -- the topic model measures the frequency of features/OTUs within samples, so it essentially normalizing by the total number of reads. But this is done via the topic model, so no additional normalization was done before fitting the model besides normalizing by 16S copy number differences (an approach done in PICRUSt). The cited article (33) is referring to an early preprint of these results where we compared different normalization approaches used before fitting the topic model, to see if additional normalization is warranted. We compared rarefying, the DESeq2 approach, and simply dividing by the total number of reads (which again, is done implicitly by the topic model). Our results indicated that no additional normalization was necessary. We now point out that these points are detailed in the appendix (S2_appendix.docx) which is now mentioned. Also, we mentioned that we do copy number normalization for downstream PICRUSt use.

Line 681

Mulinomial  Multinomial -- done

Reviewer #3: 1. The “Themetagenomics” terminology seems rather an arbitrary measurement of the co-occurring microbial functional attributes in a microbial community without a strong scientific basis. The authors should revisit the title and consider revising. It is a “catchy” title but lacks a strong scientific basis for which the authors attempted to establish a unique bioinformatics tool. -- We like our package name, but after consideration of this comment, we are dropping themetagenomics from the title.

2. The analysis is based upon the HTS of 16S rRNA gene amplicons within the microbial metacommunity DNA to determine predictive metabolic function like the PiCrust analysis of a co-occurring subset of the microbial community (based on the CoNet analysis). The CoNet analysis may produce highly variable keystone taxa including the nodes on the same HTS sample. Thus the reproducibility of the CoNet results may not be achieved. -- Functional prediction from taxonomic abundances faces many challenges. One advantage of using topics instead of co-occurrence networks is that each topic consists of estimated relative frequencies of OTUs, which allows for downstream approaches such as using PICRUSt to gain insight into functional content differences among topics. Also, topics need not be “pruned” to find distinct groups of co-occurring taxa; the topics themselves are the groups. Also, given that the topic model presents relative frequencies of OTUs within topics and relative frequencies of topics within samples, these frequencies can easily be used to identify associations with sample-level information via familiar methods such as regression. Lastly, the structural topic model can be given an initialization (or initial seed) that yields the same results every time -- therefore reproducibility is ensured and the taxa-to-topic mapping will not change with the same initializations.

3. How do the microbial functional attributes and co-occurring microbial community from 16S rRNA amplicon HTS data if conducted compares to the shotgun metagenomics? Would microbial taxa (OTUs) be different between the two approaches? Validation is necessary. -- We address this in the section titled “Validating the Functional Predictions of Themetagenomics via Paired MGS Samples.” We showed that the predicted functional content between sample matched 16S rRNA amplicon surveys matched the functional content found in the same topics using metagenomic shotgun sequencing data. The purpose of this simulation was to show that, despite multiple steps, the signal can be trusted. We are fitting topics with 16S rRNA gene amplicon surveys, then using PICRUSt to predict within-topic functional content, and then fitting a fully Bayesian multi-level model to extract topic-gene pathway interactions. We were concerned about spurious correlations/false positives given the number of steps. Hence, we validated our approach using MGS data.

In terms of comparing topic/OTU frequency using 16S rRNA amplicon surveys and topic/taxa frequencies using MGS data, we don’t believe this will provide much beyond showing that the two datasets are consistent, which is done in the paper describing the data we used. Topic models have been used for quite some time and have been shown to accurately estimate feature co-occurrence. Given our simulations in the supplement, previous topic model literature, and topic models used in microbiome work, there is a strong foundation that shows the “clustering” is accurate. Comparing topics between the two datasets will simply show that the datasets themselves are similar, which is described in the paper that generated the data.

4. The primary objective of this study seems the development of a new and unique bioinformatics tool to precisely identify the co-occurring microbial sub-community and the keystone taxa and then use them to determine the predictive functional attributes. The key issue with this manuscript is that the approach uses bias when pre-selecting the functions. If that is correct then how this approach would address the outcome of unknown sample properties during double-blinded studies? There seem to be a flaw in this approach, particularly when unknown samples are tested. The authors should address this limitation. -- We don’t understand the reviewer when s/he states that “bias” was used to “pre-select” function. Although we did omit pathway categories with little-to-no applicability to microbes, we did not influence the way in which our topics “cluster,” nor did we influence which gene functions fell into which topics. Again, we omitted categories that the genes can be annotated with. We never manipulated in anyway which gene fell into which topic. A given gene may have 5 potential pathways it can be annotated with. If one of those pathways was a human CNS pathway, we omitted it. This was done for computational concerns. More pathways led to more combinations and hence a larger feature space to work with for the Bayesian multi-level model, which are already computationally expensive.

We are also confused about where double blinding and unknown sample properties come into play. If a given dataset had no sample level information, then one would simply fit topics without any associated co-variates and explore taxa or gene function co-occurrence. S/he would not be able to link these topics to sample-level information because that information is not available, but this would not render the model obsolete, nor would one not be able to fit the model. Still, it is certainly common for a microbiome study to have sample-level information.

In terms of “unknown” samples. We are unsure what the author is suggesting. Samples without labels for training? The topic model is completely unsupervised. It does not need labels to be fitted, nor does it perform prediction, so cross validation and the like is not necessary. If one were to have “completely unknown samples,” that person can simply fit the topic model, predict the gene function, and explore the co-occurrence. Labels are not necessary, but if they are available, then additional exploration can be done to identify interesting sample-level/topic relationships, similar to any analyses that can be done on any given microbiome dataset. 

5. The study needs to use more than one sample type (multiple clinical as well as environmental) in order to establish a broadly applicable bioinformatics tool. Moreover, the authors used the HTS files available on the open-access database. However, for this study, the authors may consider designing experiments with variable sample parameters to demonstate the modulation of the co-occurring microbial community as well as the keystone taxa resulting into he changes in microbial metabolic functional attributes. -- We appreciate the suggestion by the author. In terms of “more than one sample type,” our work uses multiple human microbiome studies given their sample depth and sample-level data that’s available. These studies are well cited, well understood, and often used as datasets for methods development. So they act as a good baseline for us to interpret our results. The sample “types” are the same, however. They are all microbiome count data generated via 16S rRNA amplicon sequencing, and they all come from human subjects. 

Also, in the supplement, we generate multiple synthetic count datasets that demonstrates different modulation configurations of the co-occurring microbial communities. We vary characteristics ranging from sparsity to sample-level feature effect size of the communities. These simulations thoroughly test the topic models capability of capturing co-occurring taxa under various co-occurrence scenarios, and the results should be applicable to potential datasets from a variety of situations.

---

## [Decision Letter · Decision Letter 1]

21 Oct 2019

Exploring Thematic Structure and Predicted Functionality of 16S rRNA Amplicon Data

PONE-D-19-17255R1

Dear Dr. Woloszynek,

We are pleased to inform you that your manuscript has been judged scientifically suitable for publication and will be formally accepted for publication once it complies with all outstanding technical requirements.

With kind regards,

Juan J Loor

Academic Editor

PLOS ONE

Additional Editor Comments (optional):

Reviewers' comments:

Reviewer's Responses to Questions

**Comments to the Author**

1. If the authors have adequately addressed your comments raised in a previous round of review and you feel that this manuscript is now acceptable for publication, you may indicate that here to bypass the “Comments to the Author” section, enter your conflict of interest statement in the “Confidential to Editor” section, and submit your "Accept" recommendation.

Reviewer #1: All comments have been addressed

Reviewer #2: All comments have been addressed

2. Is the manuscript technically sound, and do the data support the conclusions?

Reviewer #1: Yes

Reviewer #2: Yes

3. Has the statistical analysis been performed appropriately and rigorously? 

Reviewer #1: Yes

Reviewer #2: Yes

4. Have the authors made all data underlying the findings in their manuscript fully available?

Reviewer #1: Yes

Reviewer #2: Yes

5. Is the manuscript presented in an intelligible fashion and written in standard English?

Reviewer #1: Yes

Reviewer #2: Yes

6. Review Comments to the Author

Reviewer #1: (No Response)

Reviewer #2: (No Response)

7. PLOS authors have the option to publish the peer review history of their article (what does this mean?). If published, this will include your full peer review and any attached files.

Reviewer #1: No

Reviewer #2: No

---

## [Editor Report · Acceptance letter]

25 Nov 2019

PONE-D-19-17255R1 

Exploring Thematic Structure and Predicted Functionality of 16S rRNA Amplicon Data 

Dear Dr. Woloszynek:

I am pleased to inform you that your manuscript has been deemed suitable for publication in PLOS ONE. Congratulations! Your manuscript is now with our production department. 

With kind regards,

on behalf of

Dr. Juan J Loor 

Academic Editor

PLOS ONE